# Privacy-Preserved Approximate Classification Based on Homomorphic Encryption

**Xiaodong Xiao, Ting Wu \*, Yuanfang Chen and Xingyue Fan**

College of Cybersecurity, Hangzhou Dianzi University, Hangzhou 310018, China; 171270014@hdu.edu.cn (X.X.); chenyuanfang@hdu.edu.cn (Y.C.); fxingyue@126.com (X.F.)

**\*** Correspondence: wuting@hdu.edu.cn

**Abstract:** Privacy is a crucial issue for outsourcing computation, which means that clients utilize cloud infrastructure to perform online prediction without disclosing sensitive information. Homomorphic encryption (HE) is one of the promising cryptographic tools resolving privacy issue in this scenario. However, a bottleneck in application of HE is relatively high computational overhead. In this paper, we study the privacy-preserving classification problem. To this end, we propose a novel privacy-preserved approximate classification algorithm. It exploits a set of decision trees to reduce computational complexity during homomorphic evaluation computation formula, the time complexity of evaluating a polynomial is degraded from $O(n)$ to $O(\log n)$. As a result, for an MNIST dataset, the Micro-$f1$ score of the proposed algorithm is 0.882, compared with 0.912 of the standard method. For the Credit dataset, the algorithm achieves 0.601 compared with 0.613 of the method. These results show that our algorithm is feasible and practical in real world problems.

**Keywords:** privacy; homomorphic encryption; machine learning; gradient boosting decision tree

## 1. Introduction

Machine learning is widely used for a variety of problems due to its attractive ability resolving real world problems. Among various machine learning techniques, Gradient Boosting Decision Tree (GBDT) is commonly used in machine learning task because of its efficiency and accuracy. It can be applied in many machine learning tasks such as multi-class classification [1], regression [2], and learning to rank [3]. Machine Learning as a Service (MLaaS) is a novel paradigm in which computing service providers make online predictions for clients.

However, privacy is an important issue in this paradigm. MLaaS requires trust between service provider and client. However, this demand is not always satisfied in real world problems. Clients may be unwilling to disclose their sensitive information, for example, assisting in medical diagnoses or detecting fraud from personal finance data. Several methodologies for privacy preserving are used such as anonymization, perturbation, randomization, and condensation. For more details, see [4].

Homomorphic Encryption (HE) is an appropriate cryptographic tool resolving privacy issues in MLaaS. HE allows computation on encrypted data without decrypt it. Due to this attractive functionality, HE has received much attention recently for preserving sensitive information (e.g., financial data). Although there also exist other cryptographic tools such as secure multiparty computation (MPC), HE has relative advantages compared to MPC since it supports no-interactive operation and fits perfectly in matrix and vector operations [5].

However, a major bottleneck of HE in applications is relatively huge computational overhead. In terms of HE, a computation is a function seen as a circuit. Therefore, the complexity for homomorphic computation is measured by the depth of the circuit. Several works are proposed to reduce computational overhead. For example, Cheon et al. [5] proposed an ensemble method

for logistic regreesion based on HE, which resulted in substantial improvement on the performance of logistic regression. Zhang et al. [6] proposed GELU-Net, which was a novel privacy-preserving architecture where clients can collaboratively train a deep neural network model. Their experiments demonstrated the stability in training and time speed-up without accuracy loss. However, huge computational overhead is an inherent shortcoming of those methods. The depth of the circuit is the most significant parameter of HE when computing a circuit for machine learning. However, most commonly used methods for evaluating a decision tree in a privacy-preserved constraint are linear algorithms. In particular, the depth of a circuit for machine learning grows linearly to the number of tree nodes. In other words, there exists a limitation on reducing the computational overhead based on HE in privacy preserving machine learning using a decision tree algorithm. To this end, we need to develop a new method, which takes full advantage of the tree algorithms' ability and reduces the computational cost of the homomorphic evaluation of the circuit.

Due to privacy and efficiency concerns, we propose a novel HE-based approximate GBDT algorithm based on the following two observations:

- When we ensemble a set of decision trees to produce online prediction, we obtain a linear algorithm but shallower circuit depth. In other words, a GBDT algorithm achieves more accurate prediction but much smaller computational overhead compared with a traditional decision tree algorithm.
- Due to many machine learning tasks being based on datasets with error, an approximate GBDT algorithm also works in real world problems if we can carefully investigate the error bound of an algorithm,

which degrades the depth of the circuit via ensembling a set of decision trees. As far as we know, this work is the first exploration to utilize the nature of the ensemble in GBDT to improve computational efficiency. As a side contribution, we also increase the stability of classification on encrypted data. Our contributions are summarized as follows:

- Chebyshev approximation of the parameterized sigmoid function. We propose a new algorithm called homomorphic approximate GBDT, which enables performing approximate classification on encrypted data using Chebyshev approximation of the parameterized sigmoid function.
- Significant reduction in computation time. We propose an improved strategy for evaluating polynomials corresponding to the decision tree. Our novel method requires $O(\log n)$ homomorphic multiplication to compute an approximate output of a decision tree, compared to $O(n)$ of the naive method, where $n$ is denoted as the number of attribute of clients' data.
- Vertical packing. In contrast to a horizontal packing technique that packs several data instances into a single plaintext, we apply a vertical packing technique to reduce space overhead substantially. We argue that a vertical packing technique is more scalable and feasible than horizontal packing in practice.

The paper is organized as follows. Section 2 introduces some backgrounds on homomorphic encryption and a GBDT algorithm. Section 3 explains how to homomorphically evaluate a pre-trained decision tree and provides an optimization algorithm to reduce the computational complexity. The homomorphic approximate GBDT algorithm is presented in Section 4. We perform experiments and discussion in Section 5. We evaluate our method and give some comprehensive robustness tests about it. We conclude the paper and identify several future works in Section 6.

## 2. Preliminaries

The rest of this section introduces some basic background that is useful in the rest of paper.

## 2.1. Problem Statement

In this work, we explore a privacy preserving prediction problem illustrated in Figure 1. It is made up of two parties, and the whole process can be described as follows: firstly, clients (e.g., institution or individuals) encrypt their sensitive data and send it to a computing service provider. A service provider (e.g., Alibaba, Tencent, or Amazon) makes predictions on encrypted data using a pre-trained machine learning model.

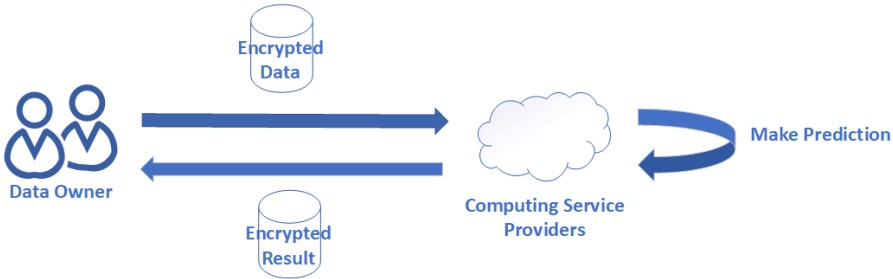

**Figure 1.** Privacy-preserving scenario.

## 2.2. Error Bound of Multivariate Function

For a multivariate function $G(a_1, a_2, \ldots, a_n)$, we denote the approximate value of $a_i$ as $\tilde{a}_i$, the approximate value of $G$ as $\widetilde{G} = \widetilde{G}(\tilde{a}_1, \tilde{a}_2 \ldots, \tilde{a}_n)$. We use $e(G) = G - \widetilde{G}$ to indicate absolute error between $G$ and $\widetilde{G}$, $\varepsilon(G) = \frac{e(G)}{\widetilde{G}}$ indicate the relative error between $G$ and $\widetilde{G}$, respectively. According to *taylor theorem*, it is easy to see that

$$e(G) = G - \widetilde{G} = \sum_{i=1}^{n} \frac{\partial G}{\partial a_i}(a_i - \tilde{a}_i) + R, \tag{1}$$

where $R = \frac{1}{2!}\left(\sum_{i=1}^{n}(a - \tilde{a}_i)\frac{\partial}{\partial a_i}\right)^2 G(\xi_1, \cdots, \xi_n)$ is the truncation error, and $\xi_i$ is a real number between $a_i$ and $\tilde{a}_i$. Hence,

**Lemma 1.** *(Absolute Error Bound) [7] $e(G)$, the absolute error bound of $\widetilde{G}$, satisfies*

$$\begin{aligned}
|e(G)| = \left|G - \widetilde{G}\right| &\leqslant \sum_{i=1}^{n}\left|\frac{\partial G}{\partial a_i}\right| \cdot |a_i - \tilde{a}_i| \\
&= \sum_{i=1}^{n}\left|\frac{\partial G}{\partial a_i}\right| \cdot \delta(a_i),
\end{aligned} \tag{2}$$

*where $\delta(a_i) \geqslant |e(a_i)|$ is the absolute error bound of $\tilde{a}_i$.*

**Lemma 2.** *(Relative Error Bound) [7] $\varepsilon(G)$, the relative error bound of $\widetilde{G}$, satisties*

$$|\varepsilon(G)| = \frac{|e(G)|}{\left|\widetilde{G}\right|} \leqslant \sum_{i=1}^{n}\left|\frac{\tilde{a}_i}{\widetilde{G}}\right| \cdot \left|\frac{\partial G}{\partial a_i}\right| \cdot \Delta(a_i), \tag{3}$$

*where $\Delta(a_i) \geqslant |\varepsilon(a_i)|$ is the relative error bound of $\tilde{a}_i$.*

Given a data instance $\mathbf{x} = (x_1, x_2, \cdots, x_n)$, the output of GBDT can be treated as a multivariate function with respect to $x_1, x_2, \cdots, x_n$. Thus, the error bound of the proposed approximate algorithm can be established by Lemma 1 and Lemma 2. It is demonstrated in Section 4 for more details.

### 2.3. Homomorphic Encryption

Homomorphic encryption is a powerful cryptographic tool, which can solve security and privacy issues in outsourcing computation. The first construction of fully homomorphic encryption is due to Gentry's excellent work [8] in 2009. In 2017, Chen et al. [9] proposed a novel scheme called Cheon–Kim–Kim–Song (CKKS) scheme which supports arithmetics of approximate numbers. Let $m$, $\mathbf{ct}$, $\mathbf{sk}$ represent the CKKS plaintext, ciphertext, and secret key, respectively. Then, the decryption algorithm is done as

$$Dec\,(\mathbf{sk}, \mathbf{ct}) = m + e \approx m, \tag{4}$$

where $e$ is a negligible error. This approximate concept of CKKS makes sense due to most of the data in real world applications being noisiness. In this setting, the CKKS scheme works perfectly in practice [5]. Using the CKKS scheme, many works addressed real world problems. For example, Cheon et al. [5] proposed an ensemble method for logistic regression based on the CKKS scheme, Zhang et al. [10] presented a practical solution for secure outsourced matrix computation and then provided a novel framework for secure evaluation of encrypted neural networks on encrypted data. All of these works demonstrated the applicability of the CKKS scheme. Based on the advantages explained above, the CKKS scheme is the underlying encryption scheme of the presented algorithm in the paper. The brief description of the CKKS scheme is as follows:

---

**CKKS homomorphic encryption scheme**

$\mathcal{R}$ is the ring $\mathbb{Z}[\mathbf{X}]\,/\,(\mathbf{X}^N + 1)$ of integer polynomials modulo $\mathbf{X}^N + 1$. We use $\mathcal{R}_q$ to denote $\mathbb{Z}[\mathbf{X}]\,/\,(\mathbf{X}^N + 1)$ with integer coefficients modulo $q$. We use $[x]_q$ to denote $x \bmod q$. The inner product between vectors $\mathbf{a}$ and $\mathbf{b}$ is denoted by $\langle \mathbf{a}, \mathbf{b} \rangle$.

**Keygen** $(1^\lambda)$ : Given the security parameter $\lambda$, select integer $p, L$, set $q_\ell = p^\ell$, where $\ell = 1, 2, \cdots, L$. Generate $\mathbf{sk}, \mathbf{pk}, \mathbf{evk}$ and then output it.

**Encrypt** $(\mathbf{m} \in \mathcal{R})$: Let $v \xleftarrow{\$} \chi_{enc}$ and $e_0, e_1 \xleftarrow{\$} \chi_{err}$. Output $[v \cdot \mathbf{pk} + (\mathbf{m} + e_0, e_1)]_{q_L} \in \mathcal{R}_{q_L}^2$.

**Decrypt** $\left( \mathbf{ct} \in \mathcal{R}_{q_\ell}^2; \mathbf{sk} \right)$ : Output $m = [\langle \mathbf{ct}, \mathbf{sk} \rangle]_{q_\ell}$.

**Add** $\left( \mathbf{ct}_1 \in \mathcal{R}_{q_\ell}^2, \mathbf{ct}_2 \in \mathcal{R}_{q_\ell}^2 \right)$ : Output $\mathbf{ct_{add}} = [\mathbf{ct}_1 + \mathbf{ct}_2]_{q_\ell}$.

**Mult** $\left( \mathbf{ct}_1 \in \mathcal{R}_{q_\ell}^2, \mathbf{ct}_2 \in \mathcal{R}_{q_\ell}^2 \right)$: Let $\mathbf{ct}_1 = (a_1, b_1), \mathbf{ct}_2 = (a_2, b_2)$, compute $(d_0, d_1, d_2) = [(a_1 a_2, a_1 b_2 + a_2 b_1, b_1 b_2)]_{q_\ell}$, Output $\mathbf{ct}_{mult} = \left[ (d_0, d_1) + \left\lfloor q_L^{-1} \cdot d_2 \cdot \mathbf{evk} \right\rceil \right]_{q_\ell}$.

**CMult** $\left( a \in \mathcal{R}, \mathbf{ct} \in \mathcal{R}_{q_\ell}^2 \right)$ : Output $\mathbf{ct}_{cmult} = \left[ (d_0, d_1) + \left\lfloor q_L^{-1} \cdot d_2 \cdot \mathbf{evk} \right\rceil \right]_{q_\ell}$.

---

### 2.4. Gradient Boosting Decision Tree

Gradient Boosting Decision Tree (GBDT) is an ensemble method of decision trees, which is trained iteration by iteration [11]. It learns a decision tree by fitting the negative gradients in each iteration [12]. Note that this paper only focuses on the inference process of GBDT, so we omit GBDT training algorithm. The significant part in GBDT lies in the evaluation of decision trees, which is the most time-consuming part in the whole process. Suppose we have a pre-trained decision tree. Given a unlabeled data instance $\mathbf{x} = (x_1, x_2, \cdots, x_n)$, where $x_i$ is defined as the $i$-th attribute in $\mathbf{x}$, we compare the attribute $x_i$ with the threshold $s_i$ allocated to internal node. The comparison result is indicated by a boolean variable

$$a_i = \mathbf{sign}\,(s_i - x_i) = \begin{cases} 1 & \text{if } s_i \geqslant x_i \\ 0 & \text{otherwise} \end{cases}. \tag{5}$$

Then, the value of $a_i$ determines whether $\mathbf{x}$ is assigned to the left subtree or right subtree. As a toy example, suppose we have a pre-trained decision tree as illustrated in Figure 2. For a unlabeled data instance $\mathbf{x} = (x_1, x_2) = (0.7, 0.4)$, the evaluation process is as following: because $x_2 = 0.4 < 0.6$ is

satisfied then $a_2 = \mathbf{sign}\,(0.6 - 0.4) = 1$, hence **x** goes to left branch of root node. Since $x_1 = 0.7 < 0.3$ is not satisfied, so $a_1 = \mathbf{sign}\,(0.3 - 0.7) = 0$, then **x** is assigned to label B which is the output of decision tree.

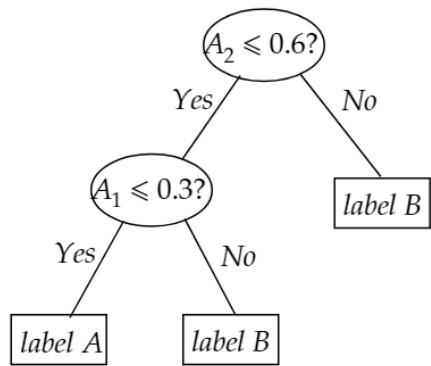

**Figure 2.** A toy example for evaluating a decision tree.

As all known HE schemes only support evaluating a polynomial homomorphically, we have to transform **sign** function into a polynomial. How to approximate a **sign** function in the context of HE is elaborated on in Section 4.

## 3. Computational Polynomial

Without loss of generality, we call the computational polynomial of a decision tree the computational model of the tree. As GBDT is an ensemble model of decision trees, the computational polynomial of GBDT is the mean value of decision trees' output. In the rest of the section, we firstly demonstrate how to transform a pre-trained decision tree to the corresponding polynomials. In addition, then we optimize the transform process. After that, we propose a GBDT computational polynomial.

### 3.1. Decision Tree Polynomial

In a binary tree, we define the internal node as the nodes that both have left children and right children. A binary tree is a true binary tree if and only if it only contains internal nodes and leaf nodes. Obviously, if $n$ is the number of internal nodes in a true binary tree, then the number of leaf of the tree is $n + 1$. Theorem 1 guarantees existence and uniqueness of polynomial form of a given decision tree.

**Theorem 1.** *Given a decision tree T that contains n number of internal nodes, then there exists only one polynomial $P\,(T)$ that can be represented as*

$$P\,(T) = \begin{cases} L_0, & \text{if } n = 0, \\ a_1 \cdot P\,(T_{left})\, , + (1 - a_1) \cdot P\,(T_{right}) & \text{otherwise,} \end{cases} \tag{6}$$

*where* $a_1 = \begin{cases} 1, & \text{if } A_1 \leqslant s_1 \\ 0, & \text{otherwise} \end{cases}$ *, and $L_0$ is the allocated label of root node.*

**Proof.** It is obvious that $n \geqslant 0$. If $n = 0$, the decision tree only contains root node, then all data instances are assigned to root nodes. Hence, $P\,(T) = L_0$ holds. If $n > 0$, then $T$ contains left subtree $T_{\text{left}}$ and right subtree $T_{\text{right}}$. As one data instance must be allocated to only one leaf node, then the label of current instance falls in $T_{\text{left}}$ or $T_{\text{right}}$, so $P\,(T) = a_1 \cdot P\,(T_{\text{left}}) + (1 - a_1) \cdot P\left(T_{\text{right}}\right)$ is satisfied. □

For simplicity, we denote $P(T)$ as $g(a_1, a_2, \cdots, a_n)$, which is a multivariate function with respect to $x_1, x_2, \cdots, x_n$. As a toy example, Figure 3 elaborates on how to transform a decision tree to a polynomial.

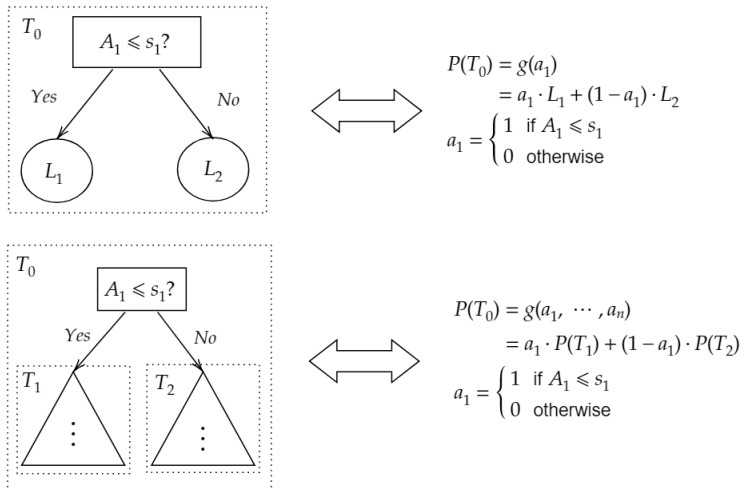

**Figure 3.** Transformation between Decision Tree and Polynomial.

Using Theorem 1, a decision tree that contained $n$ internal nodes is evaluated by $P(T) = a_1 \cdot P(T_{\text{left}}) + (1 - a_1) \cdot P(T_{\text{right}})$. The number of arithmetic operation is about $n$ additions, $n-1$ multiplication, and $n+1$ constant multiplications. Since multiplications need to perform key-switching procedure and hence is time-consuming compared to addition and constant multiplication. For this reason, we say that the complexity of Theorem 1 is $O(n)$. Since

$$
\begin{aligned}
P(T) &= a_1 \cdot P(T_{\text{left}}) + (1 - a_1) \cdot P\left(T_{\text{right}}\right) \\
&= a_1 \cdot \left(P(T_{\text{left}}) - P\left(T_{\text{right}}\right)\right) + P\left(T_{\text{right}}\right),
\end{aligned}
\tag{7}
$$

then the computational cost of decision tree evaluation is about $n-1$ additions, $\log n$ multiplications and $\approx \frac{n}{2}$ constant multiplications. Thus, the complexity of Theorem 1 is optimized to $O(\log n)$.

### 3.2. GBDT Computational Polynomial

Using the decision tree computational polynomial demonstrated above, it is straightforward to build a GBDT computational polynomial. Due to $G = \sum_{m=1}^{M} \gamma_m \cdot T_m$, firstly, we can transform each decision tree $T_m$ into the corresponding polynomial $g_m(a_1, a_2, \cdots, a_n)$; then, we average every polynomial $g_m$ and then output the expected $G = \sum_{m=1}^{M} \gamma_m \cdot g_m$.

## 4. Homomorphic Approximate GBDT Prediction

In this section, we firstly propose an algorithm for approximately evaluating a pre-trained decision tree homomorphically. Then, we present security analysis and error analysis of the proposed algorithm. In the end of the section, we propose a privacy-preserved approximate GBDT algorithm based on HE, which can be used in classification tasks on encrypted data.

### 4.1. Homomorphic Approximate GBDT Algorithm

The main cost in GBDT lies in running a decision tree algorithm for classification, and the critical part in a decision tree is comparison operation. The comparison operation is to compare the value

of attribute with the threshold in current internal node. There exists two paradigms for comparison: exact comparison and approximate comparison. For exact comparison, we can use several HE schemes such as Brakerski–Gentry–Vaikuntanathan (BGV) [13], Brakerski–Fan–Vercauteren (BFV) [14], or Gentry–Sahai–Waters (GSW) [15]. However, we have to point out that, in the setting of machine learning tasks, training data are noisy, so the trained model based on these data does not perfectly reflect the pattern. Thus, the exact comparison paradigm is not robustness enough. Fortunately, it is a judicious way to increase the robustness of the system with the help of an approximate comparison paradigm.

To realize approximate comparison, we have to approximate **sign** function. Since **sign** function is not continued and smooth, we substitute it by a parameterized sigmoid function. It is formulated as

$$\sigma(x;t) = \frac{1}{1 + \exp(-t \cdot x)}, \tag{8}$$

where $t$ is called a steep factor that controls the gap between **sign** function and parameterized sigmoid function. The larger $t$ is, the more precise an approximation function we get. The pattern is illustrated in Figure 4a. Without loss of generality, we scale $x$ to interval $[-1, 1]$. This scale operation is correct because, in classification tasks, we do not need to maintain the true values, but maintain the order between these values. In the paper, we use Chebyshev polynomials to approximate **sign** function. Figure 4b shows the approximate curves using Chebeshev polynomials. We can see that it is a very perfect curve when $degree = 24$.

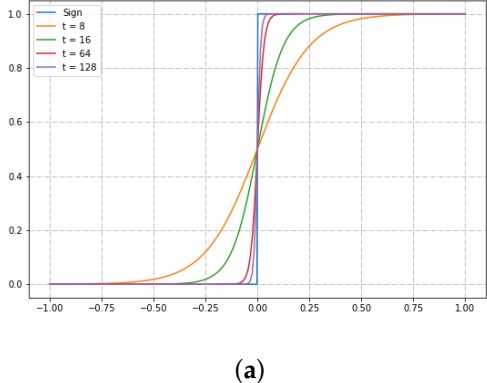

**(a)**

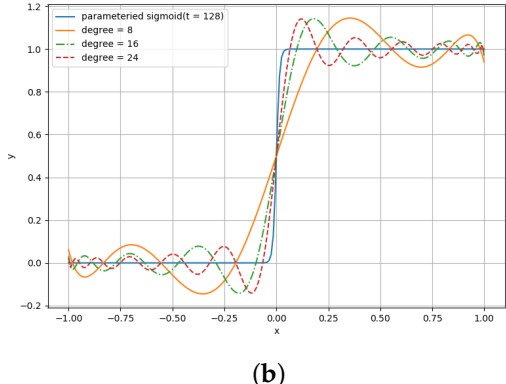

**(b)**

**Figure 4.** Parameterized sigmoid function and its approximation. (**a**) Parameterized sigmoid function; (**b**) Chebyshev approximation.

### 4.2. Security Analysis

Suppose that the cloud is honest and curious. If one can ensure the IND-CPA security of an underlying homomorphic encryption scheme (i.e., BGV scheme), then ciphertexts of any two plaintexts are computationally indistinguishable. The cloud learns nothing from the encrypted data as all computations are performed over encryption. Then, the novel algorithm we proposed preserves the confidentiality of data.

### 4.3. The Error Bound of a Decision Tree

Now, we focus on the partial derivative $a_i$, which is an internal node in $\mathbf{T}_g$. From Figure 5, we observe that

$$\frac{\partial g}{\partial a_i} = \left( P\left(T_{\text{left}}\right) - P\left(T_{\text{right}}\right) \right) \prod_{j \in \mathcal{P}} \widetilde{a}_j, \tag{9}$$

where $a_i = \begin{cases} 1, & \text{if } A_i \leqslant s_i, \\ 0, & \text{otherwise,} \end{cases}$ and $T_{\text{left}}, T_{\text{right}}$ is the left subtree and right subtree of $a_i$, respectively. $\mathcal{P}$ is the ensemble of all ancestor nodes from $a_i$ to root node in order. With maximum–minimum normalization, we can map a label attached each leaf (which is real number) into interval $[0, 1]$. Since $0 \leq P\left(T_{\text{left}}\right), P\left(T_{\text{right}}\right), \widetilde{a}_j \leq 1$, hence

$$2\frac{\partial g}{\partial a_i} \& = \left( P\left(T_{\text{left}}\right) - P\left(T_{\text{right}}\right) \right) \prod_{j \in \mathcal{P}} \widetilde{a}_j \& \leqslant \min\left(\widetilde{a}_k\right) \& \leqslant \widetilde{a}_1, \tag{10}$$

where $\widetilde{a}_k$ is a certain node in a path passed through $a_i$.

Now, we know how to express partial derivatives of each variable $a_i$. Based on it, we derive the prediction error bound of tree $T$. Suppose $\Delta = \Delta\left(a_i\right)$ is the relative error bound of internal node $a_i$ in tree $T$. With Lemma 2, we can see that $\widetilde{g}$, the final prediction of tree, $T$, satisfies

$$\begin{aligned} |\varepsilon(g)| &\leqslant \sum_{i=1}^{n} \left| \frac{\widetilde{a}_i}{\widetilde{g}} \right| \cdot \left| \frac{\partial g}{\partial a_i} \right| \cdot \Delta\left(a_i\right) \\ &= \sum_{i=1}^{n} \left| \frac{\widetilde{a}_i}{\widetilde{g}} \right| \cdot \left( P\left(g_{\text{left}}\right) - P\left(g_{\text{right}}\right) \right) \prod_{j \in \mathcal{P}} \widetilde{a}_j \cdot \Delta \\ &\leqslant \sum_{i=1}^{n} \left| \frac{\widetilde{a}_i}{\widetilde{g}} \right| \cdot \widetilde{a}_1 \cdot \Delta = \widetilde{a}_1 \cdot \Delta \cdot \frac{1}{|\widetilde{g}|} \sum_{i=1}^{n} |\widetilde{a}_i| \\ &\leqslant \widetilde{a}_1 \cdot \Delta \cdot \frac{n}{|\widetilde{g}|} \\ &\leqslant \Delta \cdot \frac{n}{|\widetilde{g}|}. \end{aligned} \tag{11}$$

Consequently, the approximate error is dominated by the following three factors:

- $\Delta$: The lower the error we want, the higher degree polynomial we have to evaluate, which leads to heavy computational time. In Section 5, we elaborate on how $\Delta$ affects the performance.
- $n$: It is a hyperparameter controlled by the service provider. A large $n$ leads to an outstanding model but is at risk of overfitting, and a small $n$ leads to lower approximate error, which is what we want. In order to obtain lower approximate error $\widetilde{g}$, it is a judicious way to increase the number of decision trees in GBDT. In this manner, we get an outstanding model that is also a lower approximate error model.
- $\widetilde{g}$: Given this parameter, we can evaluate the error bound of our approximate prediction. Note that $0 \leqslant \widetilde{g} \leqslant 1$.

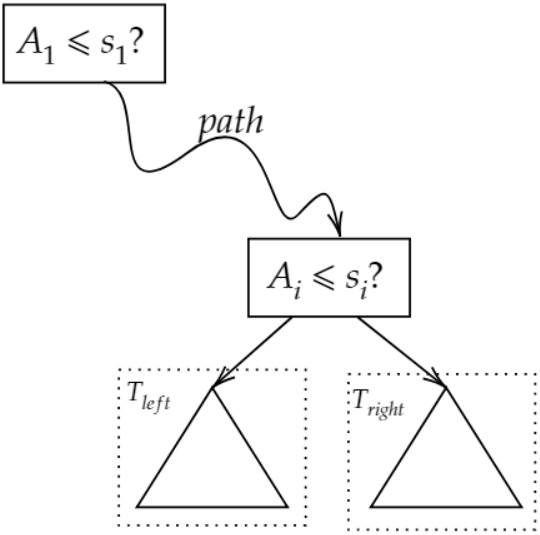

**Figure 5.** Partial derivative of an internal node.

### 4.4. The Error Bound of GBDT

As the GBDT model is an additive model of decision trees, it is easy to see that the error bound of approximate GBDT $\widetilde{G}$ is the average of decision trees error bound. To be precise, we denote a GBDT model as $G$, which consists of $M$ decision trees denoted $T_1, T_2, \cdots, T_M$. We define the prediction of $T_m$ as $g_m$, approximate prediction of decision tree as $\widetilde{g}_m$ output from Algorithm 1. Then,

$$
\begin{aligned}
|\varepsilon\left(G\right)| &= \varepsilon\left(\sum_{m=1}^{M} \gamma_m \cdot \widetilde{g}_m\right) = \sum_{m=1}^{M} \gamma_m \cdot \varepsilon\left(\widetilde{g}_i\right) \\
&\leqslant \sum_{m=1}^{M} \Delta \cdot \gamma_m \cdot \frac{n_m}{|\widetilde{g}_m|}.
\end{aligned}
\tag{12}
$$

---

**Algorithm 1:** Homo Approx. GBDT Inference

    **Input:** $\left\{(\gamma_m, T_m)\right\}_1^M, \left\{\mathbf{Ct}\left(A_i\right)\right\}_1^I$

    **Output:** $\mathbf{Ct}\left(\widetilde{G}_M\right)$

**1** Convert $T_1$ to its corresponding approximate polynomial, denoted as $\widetilde{g}_1$

**2** $\mathbf{Ct}\left(\widetilde{G}_1\right) \leftarrow \mathbf{CMult}\left(\mathbf{Ct}\left(\widetilde{g}_1\right); \gamma_1\right)$

**3** **for** $m = 2, 3, \cdots, M$ **do**

**4**      Convert $T_m$ to its corresponding approximate polynomial, denoted as $\widetilde{g}_m$

**5**      $\mathbf{Ct}\left(\widetilde{g}_m\right) \leftarrow \widetilde{g}_m\left(\left\{\mathbf{Ct}\left(A_i\right)\right\}_1^I\right)$

**6**      $\mathbf{Ct}\left(\widetilde{G}_m\right) \leftarrow \mathbf{Add}\left(\mathbf{Ct}\left(\widetilde{G}_{m-1}\right), \mathbf{CMult}\left(\mathbf{Ct}\left(\widetilde{g}_m\right); \gamma_m\right)\right)$

**7** Return $\mathbf{Ct}\left(\widetilde{G}_M\right)$

---

## 5. Experiments and Discussion

### 5.1. Experimental Settings

All experiments were implemented in C++ on Linux with AMD A6-4400M at a 2.7 GHz Processor (Advanced Micro Devices Company, Santa Clara, California, USA). The code is based on the HEAAN library [16], which is an open-source software library that implements the CKKS algorithm.

We chose MNIST [17] and the Credit [18] dataset for our experiments. These two datasets are widely used in many machine learning classification tasks. The MNIST dataset includes handwritten digits images ranging from 0 to 9. Each image is $28 \times 28$ pixels. The MNIST dataset has a training set of 60,000 samples, and a test set of 10,000 samples. The Credit dataset is a commonly used dataset for binary classification tasks. It is comprised of 28,000 samples for training and 2000 samples for tests. Each sample in the Credit dataset contains 23 attributes and one label.

In order to make a meaningful comparison of the performance of the different methods, we chose a micro $f1$-score as the performance measure. Suppose we have an $n$ different class denoted as $1, 2, \cdots, n$. For the $i$-th class, $precision_{micro}$ is computed as

$$Precision_{micro} = \frac{\sum_{i=1}^{n} TP_i}{\sum_{i=1}^{n} TP_i + FP_i},$$ (13)

and $recall_{micro}$ is computed as

$$Recall_{micro} = \frac{\sum_{i=1}^{n} TP_i}{\sum_{i=1}^{n} TP_i + FN_i},$$ (14)

where $TP_i, FP_i$, and $FN_i$ indicate true positive, false positive, and false negative with respect to $i$-th, respectively. Then, micro $f1$-score is computed as follows:

$$Micro - f1 = 2 \cdot \frac{Recall_{micro} \times Precision_{micro}}{Recall_{micro} + Precision_{micro}}.$$ (15)

Next, we provide the detailed parameters of GBDT algorithm used in our experiments. The problem that this paper study is based on an assumption that the service provider has already been training a GBDT model. Hence, we firstly run an GBDT algorithm for obtaining a pre-trained model that achieves well performance. Without a loss of generality, the parameters of the GBDT algorithm in our experiments are as follows. We chose 100 decision trees as base learners after 5-fold cross validation. For each decision tree, we set the maximum depth to 7. The learning rate is denoted by $\gamma_m$, which shrinks the contribution of each tree in each boosting stage, is set by 0.1. The $Micro - f1$ score of this parameter setting gets 0.912 on the MNIST dataset. For the Credit dataset, we train 300 trees as base learners and set the maximum depth of each tree as 3 and finally get a GBDT model with $Micro - f1 = 0.613$ on the Credit dataset. The detailed parameters of the GBDT algorithm are summarized on Table 1.

For security concern, we set $\lambda = 80, p = 2, L = 2$. Random distribution $\chi_{enc}, \chi_{err}$ is the same as [9]. The approximate error with respect to the **sign** function, $\Delta$, is $0.001, 0.01, 0.1, 0.3, 0.5$, respectively.

**Table 1.** The parameters of the GBDT algorithm.

| Datasets | Parameters | | | Performance |
|---|---|---|---|---|
| | **Learning Rate** | **# of Base Learner** | **Max Depth** | **Micro-F1 Score** |
| MNIST | 0.1 | 100 | 7 | 0.912 |
| Credit | 0.1 | 300 | 3 | 0.613 |

## 5.2. Packing Technique

A Packing Technique is a method for the acceleration of the homomorphic evaluation. More specifically, for the user's sensitive dataset $\mathbf{X} \in \mathbb{R}^{m \times n}$, each element in $\mathbf{X}$ is encrypted conventionally as an individual ciphertext. However, this method is less efficient. Packing is in general associated with the concept of batching a Single Instruction on Multiple Datasets (SIMD) [19]. Consequently, computation can be executed on each independent plaintext in a single pass.

Following the SIMD paradigm, each attribute in the same data instance can be packed as a single plaintext. Therefore, each data instance is encrypted as one ciphertext. The method is called horizontal packing in the paper. In this manner, we can see that the number of ciphertexts is the same as the amount of data instances. When the amount of a user's data is huge, the horizontal packing method has less feasibility and scalability. For instance, the Credits dataset, which is about 2.3 MB, consists of 30,000 samples. Using horizontal packing, we have 30,000 ciphertexts. For each sample, there are 23 attributes and one label, and hence it has 24 variables in each data instance. We set $\lambda$ = $2^{80}$, $q = 2^{182}$, $N = 2^{12}$; then, the size of a ciphertext is 0.355 MB. In total, the size of encrypted Credits dataset is $30,000 \times 0.355$ MB = 10.4 GB. Thus, the expansion factor is $\frac{10.4 \text{ GB}}{2.3 \text{ MB}} = 4630$, which is impractical.

We observed that there are lots of unused slots in ciphertext. In other words, it only uses $\frac{24}{2^{12}} = 0.5\%$ slots. Inspired by [4], we use a vertical packing scheme to reduce space overhead and communication complexity significantly. A vertical packing scheme packs the same attribute in the different data instances to a single plaintext. Then, each attribute in several data instances is encrypted as a ciphertext. Therefore, the number of ciphertexts is the same as the amount of attributes. Compared to the horizontal packing method, the vertical packing scheme is more feasible and scalable since the amount of ciphertexts is independent of data size. Since we have $2^{13} = 4096$ slots, a plaintext, which just contains one attribute, can be packed by 4096 data instances. Thus, for one attribute in the Credits dataset, we use $\lceil \frac{30000}{4096} \rceil = 8$ ciphertexts. For 24 attributes in the entire dataset, we use $24 \times 8 = 192$ ciphertexts. Finally, the space overhead of the encrypted dataset is $192 \times 0.355$ MB = 68.16 MB. We can see that the expansion factor is $\frac{68.16 \text{ MB}}{2.3 \text{ MB}} = 30$, which is significantly smaller than the horizontal packing method. In our experiments, we use a vertical packing technique.

### 5.3. Classification Performance on Encrypted Data

In order to analyze the influence of approximate GBDT based on HE, we first compare the performance of our algorithm to that GBDT algorithm on unencrypted data. As stated in Section 4.3, $\Delta$ is an important parameter in our algorithm. If $\Delta$ is too large, the output of our algorithm is not correct. In other words, the error bound of our algorithm can not converge if $\Delta$ is too large. We experiment several times and finally chose $\Delta = 0.001, 0.01, 0.1, 0.3, 0.5$. We argue that this choice makes sense, since it is based on the following two facts: first, if $\Delta$ is greater than 0.5, randomly choosing a classification result is better than running a privacy-preserved algorithm on the service side. Second, if $\Delta$ is too small (e.g., smaller than 0.001), the service provider must suffer much higher computational overhead for running the algorithm. A single experiment contains randomness. Consequently, we ran the experiment 10 times and plotted the median of results in Figure 6.

As shown in Table 2, a pre-trained GBDT model on MNIST achieves Micro $f_1 = 0.912$. After that, we transformed the pre-trained model to an approximate GBDT model based on the method demonstrated on Section 4. We experimentally compared the performance of approximate GBDT for various choices of $t$ and *degree* and finally selected $t = 128$, *degree* $= 24$. We can see that, when $\Delta = 0.01, 0.1, 0.3$, the output of approximate GBDT is $0.882, 0.876$, and $0.843$, respectively, while the benchmark is 0.912. Consequently, it is acceptable in our experiment when $\Delta \leq 0.3$. However, $\Delta \geq 0.3$ is not a good choice in practice because the Micro-$f_1$ score is getting much smaller compared with the performance of GBDT on unencrypted data. The results show that the approximate approach proposed in this paper is feasible and robust when we choose some appropriate parameters. For the Credit dataset, we can check that approximate error rate = 2.0%, 2.6%, 9.5% when $\Delta = 0.001, 0.01, 0.1$, respectively. Unfortunately, the approximate error ratio dropped dramatically to 0.495 while $\Delta = 0.3$. The reason is that, for the Credit dataset, we have only 23 attributes, while we have $784 = 28 \times 28$ attributes in the MNIST dataset. As stated in Section 4.3, the parameter $n$ controls the error bound of the output. A too much large $n$ leads to bad performance (e.g., Micro-$f_1 = 0.495$ in this case) and an appropriate $n$ (e.g., $n = 784$) achieves good performance (e.g., approximate error rate less than 9%). The experiments on the Credit dataset told

us that a dataset containing hundreds of attributes is an ideal dataset while using an approximated machine learning manner.

**Table 2.** Classification performance for various $\Delta$.

| Setting-A | MNIST | | | | Credits | | | |
|---|---|---|---|---|---|---|---|---|
| | Micro $F_1$ Score | | | | Micro $F_1$ Score | | | |
| $\Delta$ | Unencrypted | Encrypted | Gap | Err. Rate | Unencrypted | Encrypted | Gap | Err. Rate |
| 0.001 | 0.912 | 0.882 | 0.030 | 3.3% | 0.613 | 0.601 | 0.012 | 2.0% |
| 0.01 | 0.912 | 0.876 | 0.036 | 3.9% | 0.613 | 0.597 | 0.016 | 2.6% |
| 0.1 | 0.912 | 0.843 | 0.069 | 7.6% | 0.613 | 0.554 | 0.058 | 9.5% |
| 0.3 | 0.912 | 0.829 | 0.083 | 9.1% | 0.613 | 0.495 | 0.118 | 19.2% |
| 0.5 | 0.912 | 0.568 | 0.344 | 37.7% | 0.613 | 0.218 | 0.395 | 64.4% |

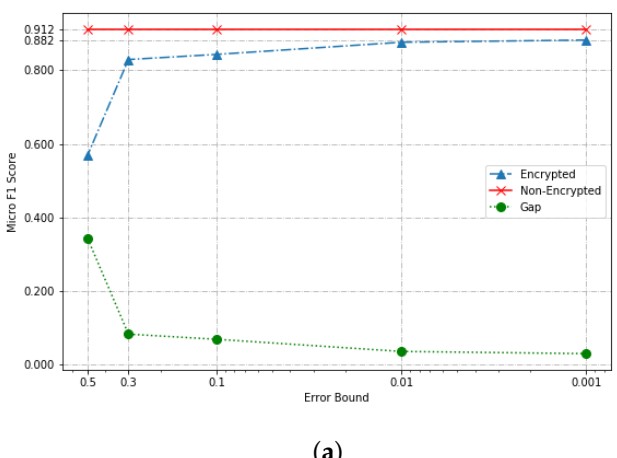

(**a**)

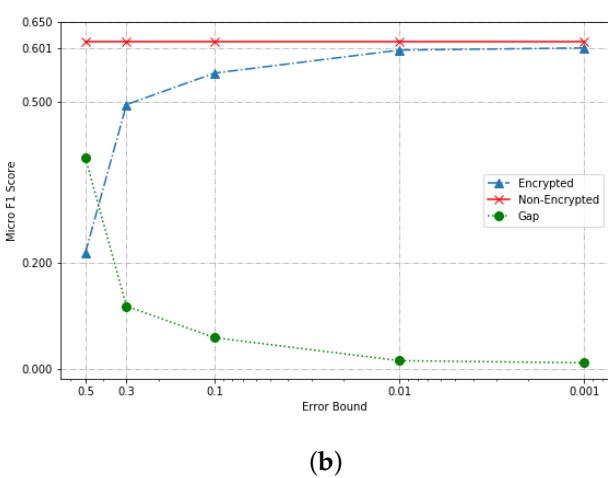

(**b**)

**Figure 6.** Performance comparison between non-encrypted and encrypted mode. (**a**) Performance on the MNIST dataset; (**b**) performance on the Credit dataset.

## 6. Conclusions and Future Work

In this paper, we proposed an approximate-based privacy preserving machine learning algorithm. It based on homomorphic encryption, which is a promising cryptographic tool resolving privacy issues. The proposed algorithm reduced the time complexity from $O(n)$ to $O(\log n)$. The proposed algorithm exploited the GBDT algorithm and CKKS schemes. Due to polynomials being an HE-friendly form

for homomorphic computation, we presented a method for how to transform a pre-trained decision tree to a polynomial. In addition, we gave a theoretical analysis on the existence and uniqueness of a polynomial form of a pre-trained decision tree. For efficiency concerns, we also proposed a refined version of a transforming procedure. Thus, the runtime complexity is optimized from $O(n)$ to $O(\log n)$. Secondly, since the polynomial approximation to the **sign** function and approximate computation of CKKS produces a small amount of errors, the theoretical analysis on the convergence of our method was demonstrated in Section 4. Then, we conducted experiments on public datasets in Section 5. From our experiments, we found that the setting of $\Delta \leq 0.3$ produces a relatively small error rate. Specifically, for the MNIST dataset, the Micro-$f_1$ score is 0.882 for encrypted data while Micro-$f_1$ score is 0.912 for unencrypted data. In addition, the error rate is 3.3%, which relatively small. Even if $\Delta = 0.3$, the error rate of the proposed algorithm is 9.1%, which is still bounded by the theoretical analysis on Section 4. For the Credit dataset, the Micro-$f_1$ score is 0.601 for encrypted data while Micro-$f_1$ score is 0.613 for unencrypted data. The error rate for Credit dataset is 2.0%, which is tolerable in practice.

The proposed algorithm is an approximate-based algorithm. However, there is no doubt that the proposed algorithm can be applied to other machine learning algorithms such as bagging (e.g., random forests) or boosting algorithms (e.g., Adaboost). Firstly, the ensemble algorithm (e.g., bagging or boosting algorithm) in machine learning using HE can significantly reduce the depth of the circuit. Compared with the non-ensemble algorithm, the HE plus ensemble algorithm is a promising solution resolving privacy issues. Secondly, it is feasible that we do approximate computation if we want to preserve the privacy information of user data. However, at the same time, we have to carefully study the approximate errors that may produce impractical results. However, the experiments in Section 5 do not exactly fit into our theoretical analysis in Section 4 because the upper bounds of approximate error are not precise enough. Although we have carefully studied the error bound, this is not enough. A better method for approximating the **sign** function may refine the error bound and finally produce a better algorithm. From our experiment results, we can see that the proposed algorithm works very well on datasets containing hundreds of attributes (e.g., MNIST dataset) but not perfectly on only a few attributes. Besides good performance of the proposed algorithm on the classification problems, generalization of the proposed algorithm would be an interesting future work for our research. In addition, finding an appropriate parameter setting would be an interesting follow-up study.

**Author Contributions:** X.X. designed, conducted formal analysis, validated the algorithm, and wrote the paper. T.W. supervised, reviewed, and edited the paper. Y.C. and X.F. reviewed and edited the paper.

**Funding:** This work was supported by the National Natural Science Foundation of China (Grant No. 61802097), the Project of Qianjiang Talent (Grant No. QJD1802020) and the Key Research Project of Zhejiang Province (No. 2017C01062).

**Conflicts of Interest:** The authors declare no conflict of interest.

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
