# Peer review of "Privacy-Preserved Approximate Classification Based on Homomorphic Encryption"

_mca, doi:10.3390/mca24040092_

Round 1

Reviewer 1 Report

Approximation based computing for homo-morphic encryption towards binary classification is proposed in this work. The complexity is significantly reduced from O(n) to O(logn) which shows a tremendous improvement. 

Paper introduces background sufficiently well.  Error bounds are seen to small with proposed technique. 

A few things to improve/address:

Introduction is pretty weak and does not introduce nor motivate the need for having this work.  It takes great effort to understand the sections 3 and 4, which are the primary contributions of this work.  Experimental results are provided, however, how does this compare with other works including PLA based approximation?  How to extend this work to other ML algorithms? And what are the parameters of the tree algorithm used?  Lots of typos in the manuscript (the deep of circuit --? the depth of the circuit, several works was ... --> several works are - and many more)

Reviewer 2 Report

Dear authors, 

I have several comments for your paper as below:

1. In Abstract, please provide brieftly quantitative results of this study, and the meaning of this work as well.
2. Introduction should be revised and rearrange, I do not prefer to put subsection like 1.1, 1.2 in Introduction part. The story mentioned in this section is not clear enough, the objective is not mentioned in the Introduction as well.
3. I suggest to separate the results section of this paper
4. I have seen no discussion in this paper, the authors show the "Experiments and discussion" section, but did not discuss about the findings of the works, and compare the findings with previous works.
5. In conclusion part, please provide brieftly quantitative results of this study.
6. n conclusion part, please provide the limitation of this study, and provide the meaning of this study. 

Round 2

Reviewer 1 Report

The authors have addressed the raised concerns. A few typos (machine learning are —> machine learning is) and citations are missing ([citation needed] exist in the manuscript. Also it will be good to include comparison if possible. 
